# Studies of prevalence: how a basic epidemiology concept has gained recognition in the COVID-19 pandemic

Diana Buitrago-Garcia [1,2] Georgia Salanti [1] Nicola Low [1]

¹Institute of Social and Preventive Medicine, University of Bern Faculty of Medicine, Bern, Switzerland
²Graduate School of Health Sciences, University of Bern, Bern, Switzerland

**Correspondence to**
Professor Nicola Low;
nicola.low@ispm.unibe.ch

## ABSTRACT

**Background** Prevalence measures the occurrence of any health condition, exposure or other factors related to health. The experience of COVID-19, a new disease caused by SARS-CoV-2, has highlighted the importance of prevalence studies, for which issues of reporting and methodology have traditionally been neglected.

**Objective** This communication highlights key issues about risks of bias in the design and conduct of prevalence studies and in reporting them, using examples about SARS-CoV-2 and COVID-19.

**Summary** The two main domains of bias in prevalence studies are those related to the study population (selection bias) and the condition or risk factor being assessed (information bias). Sources of selection bias should be considered both at the time of the invitation to take part in a study and when assessing who participates and provides valid data (respondents and non-respondents). Information bias appears when there are systematic errors affecting the accuracy and reproducibility of the measurement of the condition or risk factor. Types of information bias include misclassification, observer and recall bias. When reporting prevalence studies, clear descriptions of the target population, study population, study setting and context, and clear definitions of the condition or risk factor and its measurement are essential. Without clear reporting, the risks of bias cannot be assessed properly. Bias in the findings of prevalence studies can, however, impact decision-making and the spread of disease. The concepts discussed here can be applied to the assessment of prevalence for many other conditions.

**Conclusions** Efforts to strengthen methodological research and improve assessment of the risk of bias and the quality of reporting of studies of prevalence in all fields of research should continue beyond this pandemic.

In introductory epidemiology, students learn about prevalence, an easy to understand concept, defined as 'a proportion that measures disease occurrence of any type of health condition, exposure, or other factor related to health',[1] or 'the proportion of persons in a population who have a particular disease or attribute at a specified point in time or over a specified period.'[2] Prevalence is an important measure for assessing the magnitude of health-related conditions, and studies of prevalence are an important source of information for estimating the burden of disease, injuries and risk factors.[3] Accurate information about prevalence enables health authorities to assess the health needs of a population, to develop prevention programmes and prioritise resources to improve public health.[4] Perhaps, owing to the apparent simplicity of the concept of prevalence, methodological developments to assess the quality of reporting, the potential for bias and the synthesis of prevalence estimates in meta-analysis have been neglected,[5] when compared with the attention paid to methods relevant to evidence from randomised controlled trials and comparative observational studies.[6 7]

The COVID-19 pandemic has shown the need for epidemiological studies to describe and understand a new disease quickly but accurately.[8] Studies reporting on prevalence have been an important source of evidence to describe the prevalence of active SARS-CoV-2 infection and antibodies to SARS-CoV-2, the spectrum of SARS-CoV-2-related morbidity and helped to understand factors related to infection and disease to inform national decisions about containment measures.[9–11] Accurate estimates of prevalence of SARS-CoV-2 are crucial because they are used as an input for the estimation of other quantities, such as infection fatality ratios, which can be calculated indirectly using seroprevalence estimates.[12] Assessments of published studies have, however, highlighted methodological issues that affect study design, conduct, analysis, interpretation and reporting.[13–15] In addition, some questions about prevalence need to be addressed through systematic reviews and meta-epidemiological studies. A high proportion of published systematic reviews of prevalence, however, also have flaws in reporting and methodological quality.[5 16] Confidence in the results of systematic reviews is determined by the credibility of the primary studies and the methods used to synthesise them.

The objective of this communication is to highlight key issues about the risk of bias in studies that measure prevalence and about the quality of reporting, using examples about SARS-CoV-2 and COVID-19. We refer to prevalence at the level of a population, and not as a prediction at an individual level. The estimand is, therefore, 'what proportion of the population is positive' and not 'what is the probability this person is positive.' Although incidence and prevalence are related epidemiologically, we do not discuss incidence in this article because the study designs for measurement of the quantities differ. Bias is a systematic deviation of results or inferences from the underlying (unobserved) true values.[1] The risk of bias is a judgement about the degree to which the methods or findings of a study might underestimate or overestimate the true value in the target population,[7] in this case, the prevalence of a condition or risk factor. Quality of reporting refers to the completeness and transparency of the presentation of a research publication.[17] Risk of bias and quality of reporting are separate, but closely related, because it is only possible to assess the strengths and weaknesses of a study report if the methods and results are described adequately.

## BIAS IN PREVALENCE STUDIES

The two main domains of bias in prevalence studies are those related to the study population (selection bias) and the condition being assessed (information bias) (figure 1). Biases involved in the design, conduct and analysis of a study affect its internal validity. Selection bias also affects external validity, the extent to which findings from a specific study can be generalised to a wider, target population in time and space. There are many names given to different biases, often addressing the same concept. For this communication, we use the names and definitions published in the Dictionary of Epidemiology.[1]

## Selection bias

Selection bias relates to the representativeness of the sample used to estimate the prevalence in relation to the target population. The target population is the group of individuals to whom the findings, conclusions or inferences from a study can be generalised.[1] There are two steps in a prevalence study at which selection bias might occur: at the invitation to take part in the study and, among those invited, who takes part (figure 1).

### Selection bias in the invitation to take part in the study

The probability of being invited to take part in a study should be the same for every person in the target population. Evaluation of selection bias at this stage should, therefore, account for the complexity of the strategy for identification of participants. For example, if participants are invited from people who have previously agreed to participate in a registry or cohort, each level of invitation that has contributed to the final setting should be judged for the increasing risk of self-selection. Those who are invited to take part might be defined by demographic characteristics, for example, children below 10 years or study setting (eg, hospitalised patients), or a random sample of the general population. The least biased method to select participants in a prevalence study is to sample at random from the target population. For example, the Real-time Assessment of Community Transmission (REACT) Studies to assess the prevalence of the virus, using molecular diagnostic tests (REACT-1) and antibodies (REACT-2), invite random samples of people, stratified by area, from the National Health Service patient list in England.[9] Those invited are close to a truly random sample because almost everyone in England is registered with a general practitioner. In some cases, criteria applied to the selection of a random sample might still result in considerable bias. For example, a seroprevalence study conducted in Spain did not include care home addresses, which could have excluded around 6% of the Spanish older population.[18] Excluding people in care homes facilities might underestimate SARS-CoV-2 seroprevalence in older adults, if their risk of exposure was higher than the average in the general population.[13] Other methods of sampling are at risk of selection bias. For example, asking for volunteers through advertisements are liable to selection bias because not everyone has the same probability of seeing or replying to the advert. For example, the use of social media to invite people to a drive-in test centre to estimate the population prevalence of antibodies to SARS-CoV-2,[19] or online adverts to assess mental health symptoms during the pandemic, excludes those without an internet connection or who do not use social media, such as older people.[20]

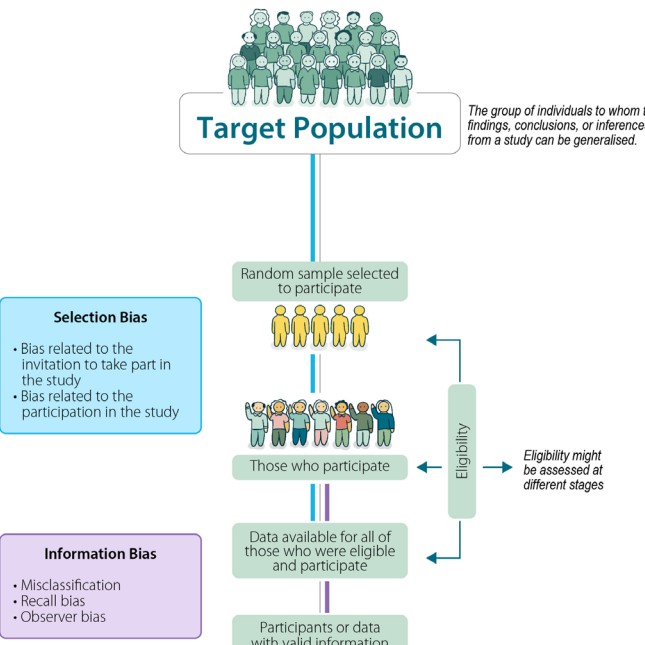

**Figure 1** Potential for selection bias and information bias in prevalence studies. Coloured lines relate to the coloured boxes, showing at which stage of study procedures selection bias (blue line) and information bias (purple line) can occur.

### Selection bias related to who takes part in the study

Non-response bias occurs when people who have been invited, but do not take part in a study differ systematically from those who take part in ways that are associated with the condition of interest.[21] In the REACT-1 study,[22] for example, across four survey rounds, the investigators invited 2.4 million people; 596 000 swabs that were returned had a valid result (25%). The proportion of participants responding was lower in later than in earlier rounds, in men than women and in younger than older age groups. If the sociodemographic characteristics of the target population are known, the observed results could be weighted statistically to represent the overall population but might still be biased by unmeasurable characteristics that drive willingness to take part.

The direction of non-response bias is often not predictable (can result in over-or underestimation of the true prevalence) because information about the motivation to take part in a study, or not, is not usually collected.[13] In a multicentre cross-sectional survey of the prevalence of PCR-determined SARS-CoV-2 in hospitals in England, the authors suggested that different selection biases could have had opposing effects.[23] For example, staff might have volunteered to take part if they were concerned that they might have been exposed to COVID-19. If such people were more likely than unexposed people to be tested, prevalence might be overestimated. Alternatively, workers in lower-paid jobs, without financial support might have been less likely to take part than those at higher grades because of the consequences for themselves or their contacts if found to be infected. If the less-well paid jobs are also associated with a higher risk of exposure to SARS-CoV-2, the prevalence in the study population would be underestimated. Accorsi *et al* suggest that the risk of non-response bias in seroprevalence studies might be reduced by sampling from established and well-characterised cohorts with high levels of participation, in whom the characteristics of non-respondents are known.[13]

As the proportion of invited people that do not take part in a study (non-respondents) increases, the probability of non-response bias might also increase if the topic of the study influences the probability and the composition of the study population.[24] Empirical evidence of bias was found in a systematic review of sexually transmitted *Chlamydia trachomatis* infection; prevalence surveys with the lowest proportion of respondents found the highest prevalence of infection, suggesting selective participation by those with a high risk of being infected.[25] Whether or not there is a dose–response relationship between the proportion of non-respondents and the likelihood of SARS-CoV-2 infection is unclear. The risks of selection bias at the stages of invitation and participation can be interrelated and might oppose each other. In the REACT-1 study,[22] it is not clear whether the reduction in selection bias through random sampling outweighed the potential for selection bias owing to the high and increasing proportion of non-respondents over time or vice versa.

### Information bias

Information bias occurs when there are systematic errors affecting the completeness or accuracy of the measurement of the condition or risk factor of interest. There are different types of information bias.

### Misclassification bias

This bias refers to the incorrect classification of a participant as having, or not having, the condition of interest. Misclassification is an important source of measurement bias in prevalence studies because diagnostic tests are imperfect and might not distinguish clearly among those with and without the condition.[26] For diagnostic tests, the predictive values will also be influenced by the prevalence of the condition in the study population. Seroprevalence studies are essential for determining the proportion of a population that has been exposed to SARS-CoV-2 up to a given time point. Detection of antibodies is affected by the test type and manufacturer, sample type such as serum, dried blood spots, saliva, urine or others,[27 28] and the time of sampling after infection. Different diagnostic tests might also be used in participants in the same study population, but adjustment for test performance is not always appropriate because the characteristics derived from studies in which the tests were validated might differ from the study population.[13] Accorsi *et al* have described in detail this issue and other biases in the ascertainment of SARS-CoV-2 seroprevalence studies.[13] Test accuracy can also change across populations, owing to the inherent characteristics of tests when clinical variability is present,[29] for example, when tests for SARS-CoV-2 detection are applied to people with or without symptoms.

In a new disease, such as COVID-19, diagnostic criteria might not be standardised or might change over time. For example, accurate assessment of the prevalence of persistent asymptomatic SARS-CoV-2 infection requires a complete list of symptoms and follow-up for a sufficiently long duration to ensure that symptoms did not develop later.[15 30] In a prevalence study conducted in a care home in March 2020, patients were asked about typical and non-typical symptoms of COVID-19. However, symptoms such as anosmia or ageusia had not been reported in association with SARS-CoV-2 at that time, so patients with these as isolated symptoms could have been wrongly classified as asymptomatic.[15 31] Poor quality of data collection has also been found in studies estimating the prevalence of mental health problems during the pandemic.[32] The use of non-validated scales, or dichotomisation to define the cases using inappropriate or unclear thresholds, will bias the estimated prevalence of the condition. Misclassification may also occur in calculations of the prevalence of SARS-CoV-2 in contacts of diagnosed cases if not all contacts are tested, and it is assumed that individuals that were not tested were also uninfected.[13]

### Recall bias

This bias results in misclassification when the condition has been measured through surveys or questionnaires

**Table 1**   Items from the STROBE checklist for cross-sectional studies that are relevant for prevalence studies

| Item | Item no | Recommendation in STROBE checklist for cross-sectional studies |
|---|---|---|
| Setting | 5 | Describe the setting, locations and relevant dates, including periods of recruitment, exposure, follow-up and data collection |
| Participants | 6a | Give the eligibility criteria, and the sources and methods of selection of participants |
| Variables | 7 | Clearly define all outcomes, exposures, predictors, potential confounders and effect modifiers. Give diagnostic criteria, if applicable |
| Data sources/ measurement | 8 | For each variable of interest, give sources of data and details of methods of assessment (measurement). Describe comparability of assessment methods if there is more than one group. |
| Statistical methods | 12c | Explain how missing data were addressed |
|  | 12d | If applicable, describe analytical methods taking account of sampling strategy |
| Participants | 13a | Report numbers of individuals at each stage of study—eg numbers potentially eligible, examined for eligibility, confirmed eligible, included in the study, completing follow-up and analysed |
|  | 13b | Give reasons for non-participation at each stage |

STROBE, Strengthening the Reporting of Observational Studies in Epidemiology.

that rely on memory. A study that aimed to describe the characteristics and symptom profile of individuals with SARS-CoV-2 infection in the USA collected information about symptoms before, and for 14 days after, being enrolled in the study.[33] The authors discuss the potential for recall bias when collecting symptoms retrospectively and if different people recollect different symptoms.

### Observer bias

This bias occurs when an observer provides a wrong measurement due to lack of training or subjectivity.[21] For example, a study in the USA found variation between 14 universities in the prevalence of clinical and subclinical myocarditis in competitive athletes with SARS-CoV-2 infection.[34] One of the diagnostic tools was cardiac magnetic resonance imaging and authors attributed some of the variability to differences in the protocols and the expertise among assessors. To reduce the risk of observer bias, researchers should aim to use tools that minimise subjectivity and standardise training procedures.

### REPORTING STUDIES OF PREVALENCE

There is no agreed list of preferred items for reporting studies of prevalence. The published article or a preprint are usually the only available record of a study to which most people, other than the investigators themselves, have access. The written report, therefore, needs to contain the information required to understand the possible biases and assess internal and external validity. The Strengthening the Reporting of Observational Studies in Epidemiology (STROBE) statement is a widely used guideline, which includes recommendations for cross-sectional studies that examine associations between an exposure and outcome.[35] Table 1 shows selected items from the STROBE statement and recommendations for cross-sectional studies that are particularly relevant to the complete and transparent description of methods for studies of prevalence.

First, clear definitions of the target population, study setting and eligibility criteria to select the study population are required (STROBE items 5, 6a). These issues

affect assessment of external validity[36] because estimates of prevalence in a specific population and setting are often generalised more widely.[1 14] Second, the denominator used to calculate the prevalence should be clearly stated, with a description of each stage of the study showing the numbers of individuals eligible, included and analysed (STROBE item 13a, b). Accurate reports of the numbers and characteristics of those who take part (responders) or do not take part (non-responders) in the study are needed for the assessment of selection bias, but this information is not always available.[24 37] Poor reporting about the proportion of responders has been described as one of the main limitations of studies in systematic reviews of prevalence.[38] As with reports of studies of any design, the statistical methods applied to provide prevalence estimates, including methods used to address missing data (STROBE item 12c) and to account for the sampling strategy (STROBE item 12d) need to be reported clearly.[35] The setting, location and periods of enrolment and data collection (STROBE item 5) are particularly important for studies of SARS-CoV-2; the stage of the pandemic, preventive measures in place and virus variants in circulation should all be described because these affect the interpretation of estimates of prevalence. Third, it is crucial to provide a clear definition of the condition or risk factor of interest (STROBE item 7) and how it was measured (STROBE item 8), so that the risk of information bias can be assessed. The definition may be straightforward if there are objective criteria for ascertainment. For example, studies of the prevalence of active SARS-CoV-2 infection should report the diagnostic test, manufacturer, sample type and criteria for a positive result.[39 40] For new conditions that have not been fully characterised, such as post-COVID-19 condition, also known as 'long COVID-19', reporting of prevalence can be challenging.[41 42] The WHO produced a case definition[43] in October 2021, but this might take time to be adopted widely.

## CONCLUSION

The COVID-19 pandemic has produced an enormous amount of research about a single disease, published over a short time period.[44 45] Authors who have assessed the body of research on COVID-19 have highlighted concerns about the risks of bias in different study designs, including studies of prevalence.[13 44] In systematic reviews of a single topic, the occurrence of asymptomatic SARS-CoV-2 infection, we observed high between-study heterogeneity, serious risks of bias and poor reporting in the measurement of prevalence.[30] Biased results from prevalence studies can have a direct impact at the levels of the individual, community, global health and policy-making. This communication describes concepts about risks of bias and provides examples that authors can apply to the assessment of prevalence for many other conditions. Future research should be conducted to investigate sources of bias in studies of prevalence and empirical evidence of their influence on estimates of prevalence. The development of a tool that can be adapted to assess the risk of bias in studies of prevalence, and an extension to the STROBE reporting guideline, specifically for studies of prevalence, would help to improve the quality of published studies of prevalence in all fields of research beyond this pandemic.

**Acknowledgements** We would like to thank Yuly Barón, who created figure 1.

**Contributors** DB-G, GS and NL conceptualised the project. DB-G and NL wrote the manuscript. GS and NL provided feedback. GS and NL supervised the research. All authors edited the manuscript and approved the final manuscript.

**Funding** This work received support from the Swiss government excellence scholarship (grant number 2019.0774), the SSPH+ Global PhD Fellowship Programme in Public Health Sciences of the Swiss School of Public Health, and the Swiss National Science Foundation (project number 176233, and National Research Programme 78 COVID-19, project number 198418).

**Competing interests** None declared.

**Provenance and peer review** Not commissioned; externally peer reviewed.

**ORCID iDs**
Diana Buitrago-Garcia http://orcid.org/0000-0001-9761-206X
Georgia Salanti http://orcid.org/0000-0002-3830-8508
Nicola Low http://orcid.org/0000-0003-4817-8986

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
