## [Reviewer comments · BMJ Open]

ARTICLE DETAILS

TITLE (PROVISIONAL)	Studies of prevalence. How a basic epidemiology concept has gained recognition in the COVID-19 pandemic.
AUTHORS	Buitrago-Garcia, Diana; Salanti, Georgia; Low, Nicola

VERSION 1 – REVIEW

REVIEWER	Hoffmann, Falk Carl von Ossietzky Universitat Oldenburg
REVIEW RETURNED	19-May-2022

GENERAL COMMENTS	The authors discuss potential problems in conducting and reporting of prevalence studies and use the example of COVID-19. Such comments are often very helpful for readers or to initiate a discussion in the community. However, it is a pity that most journals do not publish such types of articles any more. The starting point of this manuscript is clearly described. But it could benefit from some more examples of (biased) studies and clarification on consequences or lessons learned for the reader. 1. Maybe you can be clearer when you are discussing on a public health level regarding under- or overestimation of a prevalence and on an individual perspective in terms of misclassification. A prevalence estimate can be correct, but classifications are not.2. On page 6, the authors describe that selection bias can occur in two steps in prevalence studies. Often, there is an initial first step as sometimes a registry or something like this for the target population is not available. For instance, when you are interested on the prevalence in a selected population, e.g. in patients with multiple sclerosis. Maybe you also mean this with "invitation to take part", but in my opinion it would be selection bias in a sampling strategy in order to recruit a random sample of the target population.3. Do you also have published examples where persons in long-term care facilities were systematically underrepresented? In some other parts, more concrete examples from studies would also be fine, e.g. with regard to relevant (biased) studies or studies published in high impact journals with poor reporting.4. Epidemiologically spoken, a response is a proportion and not a rate (as no person time is in the denominator), although it is often named response rate.5. Misclassification might also occur when different tests (e.g. when not thousands of tests from same manufacturer is available) are used or when persons do not use the test in the same manner (e.g. when swabs are done by patients themselves or when training was not appropriate).6. A table for specific items on the STROBE checklist with regard to prevalence would be useful.
--

	7. On page 8, you state that accurate numbers and characteristics on responder and non-responder should be given. In an ideal world, this is true. But often you do not have these information. 8. The sampling strategy is also an important aspect that should be described in prevalence studies. 9. After reading the article, it would be nice to have a clear idea of what exactly are its consequences. Is it to point out that prevalence studies on COVID-19 are often biased, or that reporting can be improved in prevalence studies or that STROBE is not specific enough for this type of designs or that we need more studies on the reporting of prevalence studies? In my opinion, such comments need a strong story and clear messages. The conclusion here is somewhat vague.
--	--

REVIEWER	de man, Jeroen University of Antwerp Faculty of Medicine and Health Sciences
REVIEW RETURNED	02-Jun-2022

GENERAL COMMENTS	This piece highlights an important topic that, indeed, often is not being given sufficient attention. The piece is well structured, well written and clear and I like the practical examples to illustrate the theory. However, novelty is lacking. Basically, this has been reported repeatedly in several textbooks since decades. I am also not convinced that the pandemic was such an eye-opener for many people, hence one may question the whole set-up (and several examples also did not relate to COVID-19). Still, it may be useful to repeat these concepts in light of the pandemic and beyond. Suggestions: Regarding the title: I am not a big fan of "the science of prevalence": not sure what exactly you are referring to. + Please emphasize that it is about the recognition of the relevance of prevalence, rather than relevance, as it was relevant before. Maybe call it 'recognition'. P 5: "Some of the processes involved in the generation of a random sample might still result in bias. For example, underestimation of SARS-CoV-2 seroprevalence in older adults could occur if the sampling frame excludes people in long term care facilities, who have been at higher risk of exposure than the average in the general population." I don't see what you mean by "a process involved in the generation of random sampling" (whatever that may mean...). If random sampling is done with a reduced sampling frame, one cannot call it random sampling for a population beyond the sampling frame. I feel the authors are mixing up concepts here which makes it more confusing. P 5: "The observed results could be weighted ..." Well, this is only possible under certain conditions (eg. the demographic proportions of the target population need to be known). Please elaborate.
---

	Conclusion The conclusion is overall weak; "The experience of the COVID-19 pandemic has highlighted the importance of biases that affect the measurement of prevalence and reporting quality." Still confused if indeed this was the case, and if so, I suggest it to be better documented. "Biased results from prevalence studies can have a direct impact at the individual and community level. " Yeah, all levels I would say, what about the policy level? Global health level? Etc. "In studies of prevalence, biases affecting internal validity can also affect the external validity of findings when estimates from a study population are interpreted as being representative and generalised to the target population. " Unclear to me, authors seem to be mixing up concepts. Violating external validity does not make an internal validity bias an external validity bias. "Some questions about prevalence need to be addressed through systematic reviews and meta epidemiological studies. A high proportion of published systematic reviews of prevalence, however, also have flaws in reporting and methodological quality.⁵ 42 Confidence in the results of systematic reviews is determined by the credibility of the primary studies and the methods used to synthesise them." Not sure why this is part of your conclusion, rather bring it to the introduction if you want to keep it. "The concepts and examples provided here can be applied to any condition or outcome. " Fully disagree, please delete and also delete from the abstract. Some of the concepts, maybe, but the examples not really. "The volume of research published during the COVID-19 pandemic has put the spotlight on research methods, interpretation of the risks of bias and reporting of studies of prevalence." Is that so? Not sure. Eventually the large volume implied a lack of consideration for these methodologic issues? Consider to delete.
--	---

VERSION 1 – AUTHOR RESPONSE

Table of comments and responses.

Comment	Author's response
Reviewer 1: Prof. Falk Hoffmann, Carl von Ossietzky Universität Oldenburg	
1. Maybe you can be clearer when you are discussing on a public health level	Thank you. We have explained this in the introductory section (p4), "We refer to

regarding under- or overestimation of a prevalence and on an individual perspective in terms of misclassification. A prevalence estimate can be correct, but classifications are not.	prevalence at a population level, and not as a prediction at an individual level. The estimand is “what proportion of the population is positive” and not “what is the probability this person is positive” P4
2. On page 6, the authors describe that selection bias can occur in two steps in prevalence studies. Often, there is an initial first step as sometimes a registry or something like this for the target population is not available. For instance, when you are interested on the prevalence in a selected population, e.g. in patients with multiple sclerosis. Maybe you also mean this with "invitation to take part", but in my opinion it would be selection bias in a sampling strategy in order to recruit a random sample of the target population.	The reviewer is correct that the section on the invitation to take part should be taken to cover this example. At the beginning of p5, we say, “Evaluation of selection bias at this stage should therefore account for the complexity of the strategy for identification of participants. For example, if participants are invited from people who have previously agreed to participate in a registry or cohort, each level of invitation that has contributed to the final setting should be judged for the increasing risk of self-selection.”
3. Do you also have published examples where persons in long-term care facilities were systematically underrepresented? In some other parts, more concrete examples from studies would also be fine, e.g. with regard to relevant (biased) studies or studies published in high impact journals with poor reporting.	We agree that specific examples are helpful. We have included an example on p5, “For example, a seroprevalence study conducted in Spain did not include care-home addresses, which could have excluded around 6% of the Spanish older population.¹⁹ Excluding people in long term facilities might underestimate of SARS-CoV-2 seroprevalence in older adults, who have been at higher risk of exposure than the average in the general population.¹³”
4. Epidemiologically speaking, a response is a proportion and not a rate (as no person time is in the denominator), although it is often named response rate.	We agree that response rate is a misnomer. We have changed the text to say (p6), “The proportion of participants responding...”
5. Misclassification might also occur when different tests (e.g. when not thousands of tests from same manufacturer is available) are used or when persons do not use the test in the same manner (e.g. when swabs are done by patients themselves or when training was not appropriate).	Thank you for pointing this out. We added a sentence, and refer to methodological article by Accorsi for more detail (p7, Misclassification bias:), “Different diagnostic tests might also be used in participants in the same study population, but adjustment for test performance is not always appropriate because the characteristics derived from studies in which the tests were validated might differ from the study population.¹³ Accorsi et al. have described in detail this issue and other biases”
6. A table for specific items on the STROBE checklist with regard to prevalence would be useful.	Thank you for this suggestion. We have added a table with the items mentioned in

	the manuscript, and added a couple of additional relevant items (p9).
7. On page 8, you state that accurate numbers and characteristics on responder and non-responder should be given. In an ideal world, this is true. But often you do not have these information.	P10, we agree and have added to the end of the sentence, "..., but this information is not always available."
8. The sampling strategy is also an important aspect that should be described in prevalence studies.	We agree, and mention this in the section on reporting (bottom of page 9) and now also in 1 table 1 as it is included in STROBE item 6a. We also mention this in the section on selection bias (p5),
9. After reading the article, it would be nice to have a clear idea of what exactly are its consequences. Is it to point out that prevalence studies on COVID-19 are often biased, or that reporting can be improved in prevalence studies or that STROBE is not specific enough for this type of designs or that we need more studies on the reporting of prevalence studies? In my opinion, such comments need a strong story and clear messages. The conclusion here is somewhat vague.	Thank you for these comments, which give us the opportunity to strengthen our conclusions, taking into account the comments that Reviewer 1 (comments 4 to 9). We have added (p11), "Future research should be conducted to investigate sources of bias in studies of prevalence and empirical evidence of their influence on estimates of prevalence. The development of a tool that can be adapted to assess the risk of bias in studies of prevalence, and reporting guidelines for studies of prevalence would help to improve the quality of published studies of prevalence beyond this pandemic"
Reviewer 2: Dr. jeroen de man, University of Antwerp Faculty of Medicine and Health Sciences	
1. Regarding the title: I am not a big fan of "the science of prevalence": not sure what exactly you are referring to. + Please emphasize that it is about the recognition of the relevance of prevalence, rather than relevance, as it was relevant before. Maybe call it 'recognition'.	We have changed the title to "Studies of prevalence. How a basic epidemiology concept has gained recognition in the COVID-19 pandemic," to incorporate the reviewer's suggestion.

2. P 5: "Some of the processes involved in the generation of a random sample might still result in bias. For example, underestimation of SARS-CoV-2 seroprevalence in older adults could occur if the sampling frame excludes people in long term care facilities, who have been at higher risk of exposure than the average in the general population I don't see what you mean by "a process involved in the generation of random sampling" (whatever that may mean...). If random sampling is done with a reduced sampling frame, one cannot call it random sampling for a population beyond the sampling frame. I feel the authors are mixing up concepts here which makes it more confusing.	We apologise that this was not clear. We have rewritten this section, and given an example, as requested by Reviewer 1 (comment 3). The revised text reads (p5), "In some cases, criteria applied to the selection of a random sample might still result in bias. For example, a seroprevalence study conducted in Spain did not include care-home addresses, which could have excluded around 6% of the Spanish older population.¹⁹ Excluding people in long term facilities might underestimate of SARS-CoV-2 seroprevalence in older adults, who have been at higher risk of exposure than the average in the general population.¹³"
3. "The observed results could be weighted ..." Well, this is only possible under certain conditions (eg. the demographic proportions of the target population need to be known). Please elaborate.	We agree. We have revised the text to say (p6), "If the sociodemographic characteristics of the target population are known, the observed results could be weighted statistically to represent the overall population..."
The conclusion is overall weak; 4. "The experience of the COVID-19 pandemic has highlighted the importance of biases that affect the measurement of prevalence and reporting quality." Still confused if indeed this was the case, and if so, I suggest it to be better documented.	We welcome the opportunity to revise the conclusion. We offer more support for the statement highlighted by the reviewer (p10-11), "The COVID-19 pandemic has produced an enormous amount of research about a single disease, published over a short time period.⁴⁴ ⁴⁵ Authors who have assessed the body of research on COVID-19 have highlighted concerns about the risks of bias in different types of clinical research,⁴⁴ and in observational study designs, including studies of prevalence.¹³ In systematic reviews of a single topic, the occurrence of asymptomatic SARS-CoV-2 infection, we observed high between-study heterogeneity, serious risks of bias and poor reporting in the measurement of prevalence."
5. "Biased results from prevalence studies can have a direct impact at the individual and community level. " Yeah, all levels I would say, what	We have revised this sentence (p11), "Biased results from prevalence studies can have a direct impact at the levels of the individual, community, global health, and policy-making."

about the policy level? Global health level? Etc.	
6. "In studies of prevalence, biases affecting internal validity can also affect the external validity of findings when estimates from a study population are interpreted as being representative and generalised to the target population. " Unclear to me, authors seem to be mixing up concepts. Violating external validity does not make an internal validity bias an external validity bias.	We agree that this sentence was not clear and we have deleted it.
7. "Some questions about prevalence need to be addressed through systematic reviews and meta epidemiological studies. A high proportion of published systematic reviews of prevalence, however, also have flaws in reporting and methodological quality.^{5 42} Confidence in the results of systematic reviews is determined by the credibility of the primary studies and the methods used to synthesise them." Not sure why this is part of your conclusion, rather bring it to the introduction if you want to keep it.	As suggested we have moved this part to the introduction (p3, para 2).
8. "The concepts and examples provided here can be applied to any condition or outcome. " Fully disagree, please delete and also delete from the abstract. Some of the concepts, maybe, but the examples not really.	We agree that this statement was not precise. We have reworded it to say, Abstract, p2, "The concepts and examples provided here can be applied to the assessment of prevalence for many other conditions." Conclusions, p11, "This communication describes concepts about risks of bias and provides examples that authors can apply to the assessment of prevalence for many other conditions."
9. "The volume of research published during the COVID-19 pandemic has put the spotlight on research methods, interpretation of the risks of bias and reporting of studies of prevalence." Is that so? Not sure. Eventually the large volume implied a lack of consideration for these	We have deleted this sentence .

methodologic issues? Consider to delete.	
--	--

1
22.08.2022

VERSION 2 – REVIEW

REVIEWER	Hoffmann, Falk Carl von Ossietzky Universitat Oldenburg
REVIEW RETURNED	24-Aug-2022

GENERAL COMMENTS	The authors revised their manuscript and most of my points are addressed. The conclusion could refer somewhat more to STROBE.  1. On page 5 and 6, the term “response rate” is still used, please delete rate. 2. Sometimes you write about “nursing facility”, “long term facilities” or “care homes”. Maybe you can always use the same term. 3. Conclusion: In their last sentence, do the authors mean that STROBE is not specific enough for prevalence studies? The authors write that “...reporting guidelines for studies of prevalence would help to improve the quality...” without referring to STROBE. This is a bit confusing as they prominently presented STROBE and its relevant items. Here, a clearer message should be given in terms of STROBE and what should be revised there.
--

REVIEWER	de man, Jeroen University of Antwerp Faculty of Medicine and Health Sciences
REVIEW RETURNED	02-Sep-2022

GENERAL COMMENTS	Dear Authors, I think you did a great job addressing the comments. I still suggest a minor modification: “In some cases, criteria applied to the selection of a random sample might still result in bias.” suggest to reformulate: “In some cases, criteria applied to the selection of a random sample might result in considerable bias.”
---

VERSION 2 – AUTHOR RESPONSE

Dear reviewers,

Thank you very much for these additional comments. Please find below our point-by-point response.

Comment	Authors' response
Reviewer 1: Prof. Falk Hoffmann, Carl von Ossietzky Universität Oldenburg	
1. On page 5 and 6, the term “response rate” is still used, please delete rate.	We apologise for the oversight. We have replaced the term rate, with replaced words or phrases underlined, on pages 6 and 7: “Accorsi et al. suggest that the risk of non-response bias in seroprevalence studies might be reduced by sampling from established and well-characterised cohorts with high levels of participation, in whom the characteristics of non-respondents are known.¹³ As the proportion of invited people that do not take part in a study (non-respondents) increases, the probability of non-response bias might also increase if the topic of the study influences the probability and the composition of the study population.²⁴ Empirical evidence of bias was found in a systematic review of sexually transmitted Chlamydia trachomatis infection; prevalence surveys with the lowest proportion of respondents found the highest prevalence of infection, suggesting selective participation by those with a high risk of being infected.²⁵ Whether or not there is a dose-response relationship between the proportion of non-respondents and the likelihood of SARS-CoV-2 infection is unclear. The risks of selection bias at the stages of invitation and participation can be interrelated and might oppose each other. In the REACT-1 study,²² it is not clear whether the reduction in selection bias through random sampling outweighed the potential for selection bias owing to the high and increasing proportion of non-respondents over time, or vice versa.” For consistency, we also deleted the term ‘secondary attack rate’ on page 8.
2. Sometimes you write about “nursing facility”, “long term facilities” or “care homes”. Maybe you can always use the same term.	Thank you. We had used the terms given in the cited publications but have now unified the terminology and used ‘care homes’ in two places. (Page 5): “Excluding people in care homes might underestimate SARS-CoV-2 seroprevalence in older adults, who have been at higher risk of exposure than the average in the general population.” Page 7: “In a prevalence study conducted in a care home in March 2020, patients were asked about typical and non-typical symptoms of COVID-19.”

3. Conclusion: In their last sentence, do the authors mean that STROBE is not specific enough for prevalence studies? The authors write that "...reporting guidelines for studies of prevalence would help to improve the quality..." without referring to STROBE. This is a bit confusing as they prominently presented STROBE and its relevant items. Here, a clearer message should be given in terms of STROBE and what should be revised there.	We have clarified the last sentence (page 11) and added that we recommend, "an extension to the STROBE reporting guideline, specifically for studies of prevalence, would help to improve the quality of published studies of prevalence in all fields of research beyond this pandemic."
---	--

Reviewer 2: Dr. Jeroen de man, University of Antwerp Faculty of Medicine and Health Sciences

1. "In some cases, criteria applied to the selection of a random sample might still result in bias." suggest to reformulate: "In some cases, criteria applied to the selection of a random sample might result in considerable bias."	Thank you. we added the word "considerable" to this sentence (page 5). "Those invited are close to a truly random sample because almost everyone in England is registered with a general practitioner. In some cases, criteria applied to the selection of a random sample might still result in considerable bias."
---	---